# National data meets AI: Machine learning for predicting overweight/obesity among ever-married Bangladeshi women

**Suman Biswas**[ID]*, **Md. Mahamudul Islam**[ID]◉, **Nusrat Islam**[ID]◉, **Md. Abdur Rahim Mia**◉

Department of Statistics and Data Science, Islamic University, Kushtia, Bangladesh

◉ These authors contributed equally to this work.
* suman.iu09@gmail.com

## Abstract

Overweight/obesity has become a critical global health issue, as these conditions are strongly associated with elevated risk of diabetes, stroke, cardiovascular disorders, and certain types of cancer. In recent decades, Bangladesh has faced a notable rise in overweight/obesity prevalence—women are more prone to obesity than men. This study presents a comprehensive strategy for identifying risk factors and predicting overweight and obesity through machine learning (ML) classifiers among ever-married Bangladeshi women aged 15–49 years. Data from the 2017–2018 BDHS, a nationally representative survey, were examined. The data were pre-processed and subsequently balanced using the synthetic minority over-sampling technique and edited nearest neighbors (SMOTE-ENN) approach. Various feature identification techniques, including Chi-Square, LASSO, and Sequential Forward Selection, were employed to determine the key risk features. Later, permutation feature importance and SHAP analysis were employed to assess the influence of these risk factors on overweight/obesity. The classification of overweight and obesity was conducted using seven machine learning models: Support Vector Machine (SVM), Logistic Regression (LR), Random Forest (RF), K-nearest Neighbors (KNN), eXtreme Gradient Boosting (XGBoost), Categorical Boosting (CatBoost), and Multilayer Perceptron (MLP). Among the evaluated models, SVM performed best, reaching 95.79% accuracy and 97.32% precision when combined with SMOTE-ENN and hyper-parameter tuning. The study found that key factors contributing to being overweight/obese include age, division, type of residence, educational levels of both the respondent and her partner, number of children, frequency of television viewing, and wealth status; where wealth status, age, and frequency of watching television have strong influences. Therefore, integrating the balancing algorithm with the embedded feature selection strategy was effective in classifying overweight/obese women and could enhance decision-making for preventive measures in public health through timely predictions of overweight/obesity.

**Data availability statement:** All relevant data are within the manuscript and its Supporting Information files.

**Funding:** This study was financially supported by the University Grants Commission of Bangladesh, following evaluation by the competent authority of Islamic University, Kushtia-7003, Bangladesh, in the form of a grant (2024–2025) received by SB. No additional external funding was received for this study. The funders had no role in study design, data collection and analysis, decision to publish, or preparation of the manuscript.

**Competing interests:** The authors have declared that no competing interests exist.

## 1. Introduction

Overweight/obesity is characterized by the abnormal or excessive accumulation of fat that may negatively affect health [1]. More specifically, overweight/obesity is a multifactorial disease resulting from a combination of factors, including imbalanced energy expenditure and excessive-calorie intake, less physical activity, genetic and environmental influences, inadequate sleep, certain medications, health conditions, socioeconomic status, ethnicity, psychosocial stress, exposure to endocrine disruptors, and gut microbiome imbalances [2]. Obesity poses a significant health challenge as it greatly elevates the risk of diseases, including type 2 diabetes, fatty liver, hypertension, heart attack, stroke, dementia, osteoarthritis, sleep apnea, and multiple cancers, ultimately diminishing both quality of life and life expectancy [1,3,4]. In 2022, the World Health Organization (WHO) reported that 2.5 billion adults aged 18 and above were overweight, including over 890 million with obesity. In Asia, adult overweight and obesity show substantial subregional variability, with prevalence reaching up to 29% based on WHO BMI criteria [5]. Pooled estimates from 1994–2023 [6] indicate a combined prevalence of 19.3% (overweight 12.4%, obesity 6.6%) across South Asian countries, including India, Pakistan, and Bangladesh. National survey data identify the Maldives as having the highest burden, often exceeding 60%, while Bangladesh and Nepal exhibit comparatively lower but rapidly increasing trends.

Fig 1 illustrates that, in 1990, only 3.5% of adults in Bangladesh were living with overweight/obesity [7]. However, by 2022, the prevalence rose to approximately 32.5%. Consequently, overweight/obesity has become a significant public health concern in Bangladesh, affecting 23.2% of males and 41.3% of females. The Bangladesh Demographic and Health Surveys (BDHS) focus on ever-married women of reproductive age (15–49 years old), revealing a dramatic shift over recent decades. The prevalence of overweight/obesity in this group increased significantly from 9.35% in 1999 to 39.14% in 2014 [8,9], indicating that Bangladeshi women are at a comparatively higher risk than men.

On the other hand, preventing obesity is a complex endeavor, as it necessitates not only changes in physical activity and dietary habits across the population, but also coordinated efforts from the government, industry, and the scientific and medical communities to support informed lifestyle choices and reduce the prevalence of overweight [10]. Most of the existing studies have largely explored the prevalence and determinants of overweight and obesity among Bangladeshi adults using traditional data analytical approaches [11–15]. The findings also revealed that a range of factors—including older age, higher income, more education, regular TV viewing, and being unemployed in an urban setting—contribute to overweight and obesity. Machine learning (ML) models generally outperform classical statistical methods in predicting overweight and underweight status because they can capture complex non-linear relationships among demographic, behavioral, and anthropometric factors. Classical approaches such as logistic regression remain valuable for their simplicity and interpretability but often show lower predictive accuracy for multifactorial conditions like obesity [16,17]. In addition, machine learning refers to the use of computational methods to uncover intricate patterns in massive datasets,

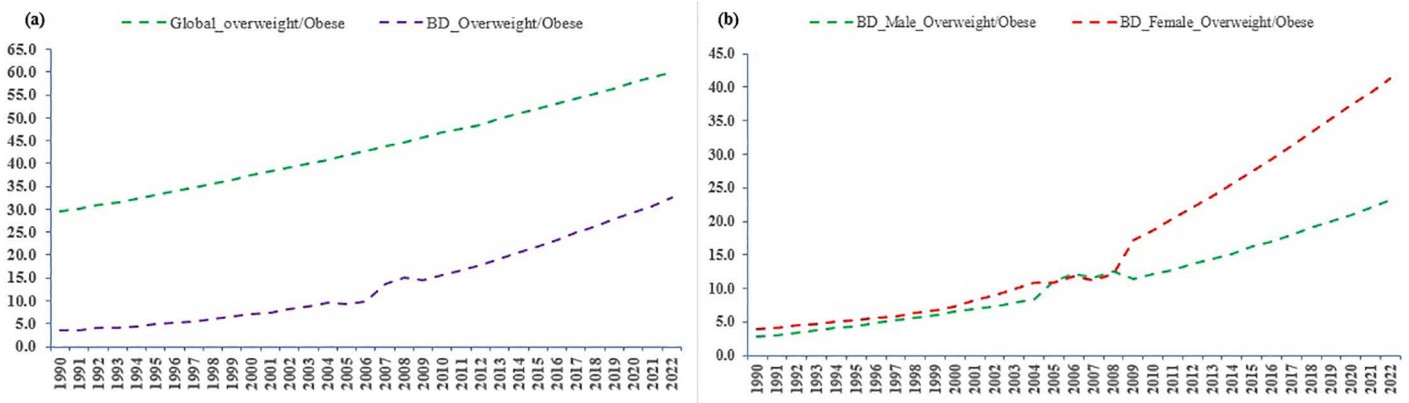

**Fig 1. Global comparison and gender-specific distribution of overweight and obesity in Bangladesh.** (a) Comparison between overweight/obesity in the World and Bangladesh; and (b) Overweight/obesity by gender in Bangladesh.

enabling the creation of accurate predictive models [18,19]. Therefore, machine learning could be a useful tool for accurately identifying the risk factors behind being overweight/obesity, which could ensure that people benefit from it easily and effectively. To the best of our knowledge, no prior research in Bangladesh has investigated the application of machine learning techniques for predicting overweight/obesity among women. However, several past and recent studies on obesity risk prediction over the world have been explored to understand the methodologies they employed and the benefits they acquired over traditional methodologies. For instance, in the study [20], the authors were able to pinpoint the major factors by integrating Shapley's additive explanation (SHAP) with machine learning approaches. They compared 7 machine learning algorithms with Embedded (penalty = "L2") feature selection methods. Their applied algorithms were LR, DT, KNN, ANN/MLP, CatBoost, RF, and GBM. The results showed that CatBoost achieved the best accuracy, at 83%. Study [20] provided valuable insights but was constrained by its single-center, cross-sectional design and limited feature set, indicating the need for broader longitudinal research. Whereas the study [21] used Logistic regression, Naïve Bayes and CART algorithms to determine the factors of obesity. They applied the Filter Method (Chi-square) as a Feature selection technique as well as the synthetic minority oversampling technique (SMOTE) to deal with data imbalance. The results showed that the LR approach achieved the best results with 72% accuracy, 71% specificity, and 69% precision. Study [21] was restricted by traditional regression methods that rely on linearity and independent assumptions and use only a small number of predictors. Likewise, the study [22] focused on developing a logistic binary classification method with the LASSO framework for accurately identifying risk factors for childhood obesity. The performance of the best prediction model was characterized by 74% accuracy, 76% sensitivity, and 73% specificity. Study [22] was limited by a small, geographically narrow sample and regression models that may overlook complex non-linear relationships.

Moreover, the study [23] utilized a trained neural network model to predict obesity levels. To determine the most critical factors associated with obesity, the researchers employed chi-square, F-classify, and mutual information-based feature selection techniques. The F-Classify approach showed the best performance, with 90.32% accuracy, 89.84% specificity, and 90.27% sensitivity. Study [23] required more diverse clinical and demographic data, balanced obesity categories, and external validation to strengthen generalizability. The study [24] proposed a Bagging-based Feature Selection framework integrated with MapReduce (BFSMR) to identify obesity risk factors. Their method was benchmarked against several others, including Mutual Information (Filter), SVM-RFE (Wrapper), LASSO and Ridge (Embedded), and the Random Forest model. The results show that Lasso and Ridge performed relatively better, with 84.4% and 83.9% accuracy, respectively.

Study [24] acknowledged the issues related to data sparsity, missing values, correlational interpretation, and regional specificity, which reduce the wider applicability of its findings.

Therefore, it can be concluded that the findings from these investigations may be limited or less reliable, as the performance of the proposed models was insufficient. Particularly in Bangladesh, there is a limited amount of research in this field that utilizes a machine learning strategy. Numerous algorithms demonstrate superior performance compared to others based on overall effectiveness. Better outcomes can be obtained when results from different machine learning models are evaluated and synthesized. Thus, the objectives of this study are:

* To identify key predictors and assess their contribution to explaining overweight and obesity among ever-married Bangladeshi women.

* To determine the most effective machine learning methods for predicting overweight and obesity.

* To identify and assess the most effective feature selection strategies for accurate prediction

## 2.  Materials & methods

### 2.1  Dataset descriptions

The dataset was obtained from the national representative and publicly available Bangladesh Demographic Health Survey (BDHS) 2017–2018, which can be accessed at dhsprogram.com/data/new-user-registration.cfm. The dataset used for this research comprises 16,303 observations with a total of 27 variables (features), including both binary and categorical input variables.

The outcome variable, overweight/obesity, was assessed through Body Mass Index (BMI) [15,25] as represented in Table 1.

The input features comprised socio-demographic, health & lifestyle information of ever-married Bangladeshi women aged 15 to 49 years. Table 2 provides detailed descriptions of these features, including their definitions and categories.

### 2.2  Data preprocessing

Data preprocessing ensures the creation of accurate and useful datasets by addressing errors, missing values, or other inconsistencies [26]. The initial step in the data processing workflow, known as data extraction, involved retrieving data from an SPSS (.sav) file. Initially, a total of 20,127 observations were available in the survey dataset. Upon assessment, 3,824 observations with missing values were identified and subsequently removed, resulting in a final analytical dataset comprising 16,303 complete cases. During the data screening process, no extreme or implausible outliers were detected that warranted removal or adjustment, and consequently, no imputation techniques were applied [27]. Following this, all categorical and ordinal variables were systematically transformed into numeric representations using Python to facilitate computational analysis and ensure compatibility with subsequent statistical and machine learning modeling. Age groups, region, and residence type were encoded ordinally; for example, age was coded from 1 ("15–19") to 7 ("45–49"), and wealth quintiles from 1 ("Poorest") to 5 ("Richest"). Binary variables, including education level, contraceptive use, and other yes/no indicators, were mapped as 0 ("No") and 1 ("Yes"). Other categorical variables, such as religion, occupation, decision-making, and sanitation status, were also mapped to numeric codes while preserving their original categorical distinctions.

**Table 1.  BMI was classified according to the WHO.**

| Not Overweight/Obese | *less than* 25.0 $kg/m^2$ |
|---|---|
| Overweight/Obese | *greater than or equal* 25.0 $kg/m^2$ |

**Table 2. Comprehensive descriptions of the dataset.**

| SN | Feature | Description | Categories |
|----|---------|-------------|------------|
| 01 | V013 | Age in 5-year group | 15-19, 20-24, 25-29, 30-34, 35-39, 40-44, 45-49 |
| 02 | V024 | Division | Barishal, Chittagong, Dhaka, Khulna, Mymensingh, Rajshahi, Rangpur, Sylhet |
| 03 | V025 | Type of place of residence | Urban, Rural |
| 04 | V106 | Respondent's highest educational level | No education, Primary, Secondary, Higher |
| 05 | V113 | Source of drinking water | Improved, Unimproved |
| 06 | V116 | Type of toilet facility | Improved, Unimproved |
| 07 | V119 | Household has: electricity | No, Yes |
| 08 | V121 | Household has: television | No, Yes |
| 09 | V122 | Household has: refrigerator | No, Yes |
| 10 | V123 | Household has: bicycle | No, Yes |
| 11 | V124 | Household has: motorcycle/scooter | No, Yes |
| 12 | V125 | Household has: car/truck | No, Yes |
| 13 | V130 | Religion | Islam, Hinduism, Buddhism, Christianity |
| 14 | V151 | Sex of household head | Male, Female |
| 15 | V159 | Frequency of watching television | Not at all, Less than once a week, At least once a week |
| 16 | V161 | Type of cooking fuel | Solid Fuel, Clean Fuel |
| 17 | V190 | Wealth index combined | Poorest, Poorer, Middle, Richer, Richest |
| 18 | V190A | Wealth index for urban/rural | Poorest, Poorer, Middle, Richer, Richest |
| 19 | V201 | Total children ever born | No child, One child, Two children, More than two children |
| 20 | V213 | Currently pregnant | No/Unsure, Yes |
| 21 | V312 | Current contraceptive method | Not using, Modern methods, Traditional methods |
| 22 | V404 | Currently breastfeeding | No, Yes |
| 23 | V701 | Husband/partner's education level | No education, Primary, Secondary, Higher, Don't know |
| 24 | V704 | Husband/partner's occupation | Unemployed, Agriculture, Business, Labour/Service, Job |
| 25 | V716 | Respondent's occupation | Unemployed, Agriculture, Business, Labour/Service, Job |
| 26 | V743A | Person who usually decides on the respondent's health care | Respondent alone, Respondent & husband/partner, Respondent & other person |

We then examined the distribution of the BMI outcome in the cleaned dataset, where participants were classified as either "Underweight/Normal" or "Overweight/Obese." This allowed for a clear understanding of the prevalence of each category and ensured that the outcome variable was properly formatted for subsequent statistical analyses and predictive modeling. This comprehensive preprocessing workflow ensured data consistency, maintained the inherent ordering of ordinal variables, and facilitated efficient computational processing for all downstream analyses. These steps helped improve data quality, making the analytics more effective and the decisions more reliable [28].

## 2.3 Train-test split

Following preprocessing, the dataset was divided into training and testing sets using the *train_test_split* function in Python, with 80% of the data used for training and 20% for testing. Analyses can be accessed from the GitHub link of https://github.com/BSuman01/Obesity-Prediction-among-Ever-married-Women.

## 2.4 Dataset balancing

When classification models are trained on unbalanced data, they tend to make biased predictions, which can compromise the accuracy of the results. Often, the minority class lacks sufficient instances for the model to learn effectively, making it

necessary to apply a data-balancing strategy [19,29]. Due to class imbalance in the dataset, SMOTE-ENN was applied to oversample the minority class and balance the training. This method integrates the Synthetic Minority Oversampling Technique (SMOTE), which synthesizes samples in the minority class, and Edited Nearest Neighbors (ENN), which reduces the number of samples in the majority class by removing the noise samples [30]. SMOTE-ENN was implemented with *sampling_strategy* = "auto" and *random_state* = 42. The method used the default SMOTE settings and default ENN settings as provided by the "imblearn" library in Python.

## 2.6 Hyper-parameter tuning and K-fold cross-validation

To ensure optimal model performance, hyperparameter tuning was performed using a 5-fold cross-validation approach. This approach splits the data into five groups, allowing each subset to be used for validation while the others are used for training [31].

## 2.7 Feature selection for developing a predictive model

High-dimensional data can lead to challenges like overfitting, excessive noise, and increased computational complexity in machine learning applications. Therefore, several techniques related to data mining and knowledge discovery, such as feature selection, are commonly employed to filter out irrelevant features from a dataset [32,33]. As shown in Fig 2, feature selection methods fall into three primary categories: filters, wrappers, and embedded methods [34]. In this study, the Chi-square, Sequential Forward Selection (SFS), and Lasso (L1 regularization) were used, each representing one of these broad categories.

### 2.7.1 Filter method: Chi-square.
Filter methods separate feature selection from the classification step, using a scoring system or relevance index to assess the data's characteristics without depending on any particular classifier. This helps identify general features without built-in assumptions. The chi-square test, a widely used univariate method, is favored in high-dimensional data analysis for its speed and ease of implementation [35,36]. The chi-square test statistic is given by,

$$\chi_s^2 = \sum_{i=1}^{m_s} \sum_{j=1}^{k} \frac{(O_{ij} - E_{ij})^2}{E_{ij}}, \ s = 1, ..., d,$$

where $O$ and $E$ are the observed and expected frequency, respectively, $m_s$ is the number of categories of the $s^{th}$ predictor, and $k$ is the number of classes of the target variable. The Chi-square statistic has a degree of freedom equal to $(m_s - 1)(k - 1)$ [37].

### 2.7.2 Wrapper method: Sequential forward selection.
Unlike filter methods, the wrapper approach evaluates feature subsets based on model performance, repeating the selection process until it finds the most effective combination

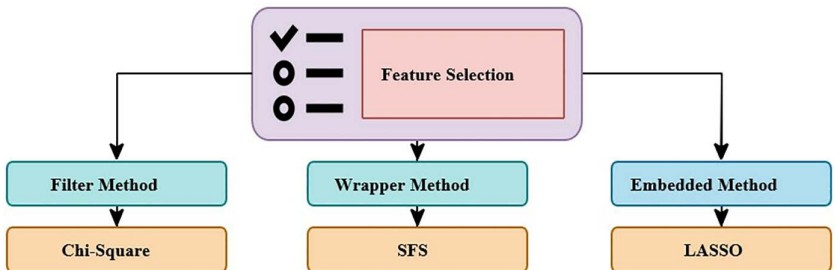

**Fig 2. Feature selection for classification.**

[38]. To identify the best set of features, wrapper methods evaluate a quality measure $Qc$ that reflects how well a chosen machine learning algorithm performs with each subset:

$$F^* = \text{argmax } Q_c\left(F^{'}\right), \qquad F' \subseteq F$$

where $C$ is any ML algorithm, and $Q$ is the quality performance of a model [36]. Sequential search methods, such as Sequential Forward Selection (SFS) and Sequential Backward Selection (SBS), explore the feature space in a unidirectional manner [39,40]. SFS, a wrapper method, starts with no features and iteratively adds those that improve performance according to predefined evaluation measures.

**2.7.3 Embedded method: LASSO (L1 regularization).** Embedded methods benefit from the properties of the Filter and Wrapper methods combined. Algorithms with inherent feature selection mechanisms facilitate their implementation [41]. This embedded method can be categorized into three, namely: pruning method, built-in mechanism, and regularization models [42]. One of the most well-known instances of the embedded technique is linear least-squares regression with the L1 regularization method, called the LASSO regularization [43]. The Least Absolute Shrinkage and Selection Operator (LASSO), introduced by Robert Tibshirani in 1996, is an effective technique that serves two primary purposes: regularization and feature selection. The primary principle of LASSO is to create a penalty function that decreases the regression coefficient of each variable to a specific range [22]. By applying an L1-norm penalty, LASSO normalizes the parameters of a linear model, reducing less correlated coefficients to zero [44]. Features with large coefficient values will be available in the chosen subsets of features. Although prior studies commonly applied L2 (Ridge) regularization, the present study employed L1 (LASSO) because it performs both coefficient shrinkage and embedded feature selection. By imposing an L1-norm penalty, LASSO forces non-informative or weakly associated predictors to zero, producing a sparse and interpretable set of features. This property was essential for the current analysis, which aimed to identify the most influential determinants of obesity within the Bangladeshi population. In contrast, L2 regularization only shrinks coefficients without eliminating variables, making it less effective for feature reduction in datasets with correlated socioeconomic predictors.

## 2.8 Feature importance

**2.8.1 Permutation.** Permutation feature importance assesses the contribution of each feature to model performance by randomly permuting its values and evaluating the resulting change in predictive accuracy. Given a trained model with a dataset and target variable, the initial performance of the model is recorded. The values of a single feature are randomly shuffled, while the values of the remaining dataset remain intact. The modified dataset is passed through the model again, and the new performance is measured. The difference between the original and new performance indicates how much the shuffled feature contributed to the prediction [45]. This process is repeated multiple times to obtain a reliable estimate of feature importance. This method can be biased when features are highly correlated, as shuffling one feature might indirectly affect another.

**2.8.2 SHAP analysis.** SHAP (SHapley Additive exPlanations) values use cooperative game theory to explain model predictions, assigning each feature an importance score by allocating the difference between the model's prediction and the average prediction to all features. Given a model $f$ trained on input data $X$, the prediction for an instance x is explained by computing SHAP values $\Phi j$, where each $\Phi j$ represents the contribution of feature j [46]. The SHAP value for feature j is calculated as:

$$\Phi = \sum_{s \subseteq F \smallsetminus \{j\}} \frac{|S|!\,(|F| - |S| - 1)!}{|F|!}\,[f(S \cup \{j\}) - f(S)]$$

where S represents a subset of features, and F is the full set of features [47]. This formulation ensures that SHAP values satisfy local accuracy, consistency, and messiness properties, making them a reliable method for feature attribution.

## 2.9 Classification algorithms

Machine learning (ML) is a rapidly evolving field that has garnered significant attention in recent years. It provides a means to extract insights from large datasets when traditional methods fall short in uncovering valuable patterns. Current research is focusing on the development of autonomous learning robots, which can adapt to dynamic environments and perform tasks without the need for explicit programming. Machine learning is being increasingly applied across various sectors, including healthcare, finance, transportation, and manufacturing [48].

**2.9.1 Support vector machine (SVM).** Support vector machine (SVM), developed by Vapnik in 1995 [49], aims to minimize classification error and maximize the geometric margin for class separation. Let us assume a set of data points consisting of n pairs $(z_1, y_1)$, $(z_2, y_{21})$, ..., $(z_r, y_r)$. Where, input vector: $z_i = (z_1, z_2, ..., z_n)$ in a real-valued space $Z \subseteq R^n$, outcome variable: $y_i$ contains a class label 1(overweight/obese) and −1 (not overweight/obese). SVM finds a linear function using the following hyperplane:

$$f(z) = w^T z + b$$

Where, w is the normalized weight vector, b is the bias of the linear classification. If the data is linearly separable, the following formula separates the data into two classes: 1 and −1.

$$y_i = \left\{ \begin{array}{ll} 1 & \text{if } w^T z + b \geq 0 \\ -1 & \text{if } w^T z + b < 0 \end{array} \right\}$$

If linear separation is challenging, the SVM kernel trick transforms the original data into a higher-dimensional space, making class separation easier.

**2.9.2 Logistic Regression (LR).** As one of the most basic yet powerful probabilistic models, Logistic Regression [50] is widely used for solving classification problems in supervised learning. It modifies Linear Regression by introducing a logistic function, making it suitable for predicting categorical outcomes. It can be defined as

$$y(x) = \sigma (\beta_0 + \beta^T x)$$

Where, $x = (x_1, x_2, ..., x)^T$ is the dataset, $\beta = (\beta_1, \beta_2, ...\beta_D)^T$ are the weights, $\beta_0$ is the bias, and σ is the logistic function.

**2.9.3 Random Forest (RF).** Random Forest (RF), developed by Breiman [45] is a robust ensemble learning algorithm applied to both classification and regression. For any test instance $z_0$, its prediction is computed through an aggregation formula determined in the initial training stage:

$$\hat{R}_{rf}(z_0) = \frac{1}{n} \sum_{i=1} \hat{r}_b(z_0)$$

**2.9.4 K-Nearest Neighbor (KNN).** K-Nearest Neighbors (KNN) is a simple classification algorithm that classifies instances based on a majority vote of their k closest neighbors in the training set. It is a type of instance-based learning, where historical data is stored and used directly for making predictions through local approximation [51]. The algorithm proceeds through the following steps: First, select the value of $K$, which represents the number of nearest neighbors to consider. Then, compute the Euclidean distance between the new instance and all instances in the training dataset using the following formula

$$d(p, q) = \sqrt{\sum_{i=1}^{n} (q_i - p_i)^2}$$

Where q represents the new instance, and $p$ is the previous instance.

**2.9.5 Extreme Gradient Boosting (XGBoost).** In the realm of data science, Extreme Gradient Boosting (XGBoost) stands out as a highly effective algorithm that applies gradient boosting to decision trees. Its efficiency and scalability make it a popular choice for solving complex, large-scale problems with minimal computational overhead [52].

To predict the results of dataset $S$ containing $p$ samples and $q$ attributes, the XGBoost model uses $m$ additive functions. Where $x_i$ is a vector in $R^q$ and $y_i$ is a real integer; this represents the dataset.
$S = \{(x_i, y_i)\}$ where, $|S| = p$, $x_i \in \mathbb{R}^q$, $y_i \in \mathbb{R}^q$.

$$\hat{y}_i = \infty (x_i) = \sum_{m=1}^{M} f_m (x_i), f_m \epsilon f(x) = w_n(x)$$

$$(\text{n} : \mathbb{R}^q \rightarrow T, w \rightarrow \mathbb{R}^T)$$

Where, $n$ represents the structure of each tree that maps a guide to the corresponding leaf nodes, and $T$ denotes the number of leaves. For every function $f_m$, there is a distinct structure of trees $n$ and leaf weights w.

**2.9.6 Categorical Boosting (CatBoost).** CatBoost [53] is an advanced gradient boosting algorithm that employs binary decision trees as base predictors and consistently delivers state-of-the-art performance across a wide range of real-world tasks. Distinct from other popular gradient boosting frameworks such as eXtreme Gradient Boosting (XGBoost) and Light Gradient Boosting Machine (LightGBM), CatBoost introduces a novel 'ordered boosting' technique to mitigate prediction shift—a form of target leakage that can adversely affect model accuracy. This ordered boosting approach modifies the standard gradient boosting process to enhance robustness and generalization in predictive modeling.

**2.9.7 Multi-Layer Perceptron (MLP).** The Multi-Layer Perceptron (MLP) classifier is a type of feedforward artificial neural network. It consists of multiple layers of nodes, where each layer is fully connected to the next adjacent layer in the network [54]. The input layer comprises nodes that represent the features of the training data. All subsequent layers transform the input data into outputs through a series of linear combinations involving input vectors, associated with weight $w$ and biases $b$. The matrix form with $k + 1$ layers can be written as:

$$y(x) = f_k(...f_2(w_2^T f1(w_1^T x + b_1) + b_2)... + b_k)$$

where y is the output, x is the input vector, w is a weight vector, and b is a bias offset.

## 2.11 Model evaluation

Although several evaluation metrics exist, a universally accepted standard for assessing the efficiency of prediction models has not yet been established [21]. This work evaluates model performance by comparing different metrics.

**2.11.1 Confusion Matrix.** It helps assess the classification model's performance, represented in a matrix that visualizes the quality for each class, as illustrated in the matrix below:

| Actual | | Predicted | | Total |
|---|---|---|---|---|
| | | **Positive** | | **Negative** |
| | Positive | TP | FP | TP+FP |
| | Negative | FN | TN | FN+TN |
| | Total | TP+FN | FP+TN | |

To assess the final predictions of the model, various classification metrics like accuracy, precision, recall, F1-score, Cohen's Kappa score, log loss, and AUC are used as described below:

$$Accuracy = \left( \frac{TP + TN}{TP + TN + FP + FN} \times 100 \right) \%$$

$$Precision = \left( \frac{TP}{TP + FP} \times 100 \right) \%$$

$$Recall = \left( \frac{TP}{TP + FN} \times 100 \right) \%$$

$$F1 - Score = 2 \times \frac{Precision \times Recall}{Precision + Recall}$$

### Cohen's Kappa Score

The Kappa score evaluates the interrelationship between the two categorical classes used for classification. It is defined as follows:

$$Kappa\ Score = \frac{p_0 - p_e}{1 - p_e}$$

where $p_0$ is the observed outcome probability, equivalent to accuracy, and $p_e$ is the probability of obtaining the desired outcome.

### Log-Loss

Log loss is a critical measure for probability-based classification models, computed as the negative average of the logarithm of the predicted probabilities, reflecting model accuracy

$$H_p(q) = -\frac{1}{n} \sum_{i=1}^{n} x_i.\log(p(x_i)) + (1 - x_i).\log(1 - p(x_i))$$

where $x$ represents the target variable level, $p(x)$ is the projected point probability for the target value, and $H(q)$ is the estimated value of log loss.

### Area Under Curve (AUC)

The Area Under the Curve (AUC) is a method for summarizing model performance, where higher values indicate better results. AUC values between 0.7 and 0.8 are good, between 0.8 and 0.9 are great, and above 0.9 are exceptional. Unlike accuracy, AUC provides a more nuanced assessment of classifier performance.

## 3. Results and discussion

After processing the BDHS 2017–18 dataset, a total sample of 16,303 ever-married women was included with 27 features. Out of the 16,303 women sampled in the study, 10,951 were underweight/normal, and 5352 were overweight/obese. Initially, a bivariate analysis was conducted on the various BMI categories among ever-married Bangladeshi women aged 15–49, using descriptive statistics expressed in terms of weighted frequencies and percentages (S1 Table). The highest share of overweight/obese respondents was found among women aged 30–34 (6.9%), and the largest proportion of respondents was found in the Chittagong division (5.6%) for overweight/obese individuals. The risk of being overweight or obese was notably higher (21.7%) among women who watched TV at least once a week compared to just 8.8% among those who watched TV less often. The majority of overweight or obese women belonged to the wealthiest households, with the highest prevalence (13.2%) observed among those with secondary or higher education in Bangladesh.

In addition, the Chi-square test was employed to assess variations in BMI categories by independent variables, with p-values $< 0.05$ considered statistically significant. Age, Division, Type of residence, Respondent's highest educational level, Husband/partner's: education level, occupation, Wealth index, Household has: electricity, television, refrigerator, motorcycle/scooter, car/truck, Type of cooking fuel, Type of toilet facility, Source of drinking water, Frequency of watching television, Total children ever born, Current contraceptive method, Currently breastfeeding, Person who usually decides on the respondent's health care were significantly associated with BMI, and those factors have large Chi-square values with $P < 0.001$.

The relationship ($\chi^2 = 649.711$, $P < 0.001$) was strong, indicating that metabolic rate declines and sedentary behavior increase with age. Regional differences across divisions ($\chi^2 = 240.024$, $P < 0.001$) likely reflect variations in urbanization, dietary patterns, and socioeconomic conditions. Urban residence ($\chi^2 = 414.680$, $P < 0.001$) is well recognized as increasing the risk of obesity due to easier access to calorie-dense foods and lower physical activity. Higher education ($\chi^2 = 231.290$, $P < 0.001$) and household assets such as electricity ($\chi^2 = 300.942$), television ($\chi^2 = 718.693$), and refrigerator ($\chi^2 = 1051.853$)—all $P < 0.001$—serve as markers of socioeconomic status; evidence from LMICs consistently shows that higher SES is linked with dietary shifts toward processed foods and more sedentary occupations.

Frequent television watching ($\chi^2 = 516.735$, $P < 0.001$) increases sedentary time and exposure to food advertisements, both associated with weight gain. Clean cooking fuel ($\chi^2 = 647.596$, $P < 0.001$) and vehicle ownership (motorcycle: $\chi^2 = 219.299$; car/truck: $\chi^2 = 39.998$, both $P < 0.001$) further indicate lifestyle modernization, which reduces physical labor. The wealth index shows the strongest associations (combined $\chi^2 = 1326.676$; urban/rural $\chi^2 = 946.133$; both $P < 0.001$), consistent with the "nutrition transition" theory, which posits that wealthier households consume more energy-dense foods. Reproductive factors—including children ever born ($\chi^2 = 133.619$, $P < 0.001$), contraceptive use ($\chi^2 = 18.057$, $P < 0.001$), breastfeeding ($\chi^2 = 344.711$, $P < 0.001$), and current pregnancy ($\chi^2 = 4.066$, $P = 0.044$)—may affect weight through biological changes during and after pregnancy, as supported by maternal health literature.

Partner-related factors, such as the husband's education ($\chi^2 = 465.805$, $P < 0.001$) and occupation ($\chi^2 = 437.689$, $P < 0.001$), reflect the household's socioeconomic position and lifestyle patterns. Health decision-making power ($\chi^2 = 50.806$, $P < 0.001$) may indicate autonomy that influences diet and health behaviors. Overall, these associations support well-established theories linking overweight/obesity to socioeconomic status, lifestyle transitions, urban environments, and reproductive history among ever-married women in Bangladesh.

### 3.1 Model results

An imbalanced class within the dataset may adversely impact the classification algorithm's performance [29]. The number of underweight/normal data is 10,951 (the majority class), and the number of overweight/obese data is 5,352 (the minority class). Thus, there is a significant difference in sample size between the two categories, resulting in an imbalanced dataset. Therefore, to enhance the performance of the machine learning models, the SMOTE-ENN method was employed

to address class imbalance, while three types of feature selection techniques—Filter, Embedded, and Wrapper—were utilized to identify the most relevant features. The amount of data categorized by the SMOTE-ENN technique (Table 3).

The SMOTE-ENN technique offers superior class balancing for the embedded method compared to filter and wrapper methods by reducing the majority class size, thereby improving overall performance, even for the default models. The performances were compared for the balanced and imbalanced to acquire significant differences in accuracy on three different feature selection techniques with seven ML algorithms: SVM, KNN, RF, XGBoost, CatBoost, MLP, and LR (Table 4).

The comparison between models with and without SMOTE-ENN demonstrates that classifiers trained on balanced datasets consistently outperform those trained on imbalanced datasets. After applying the SMOTE-ENN balancing technique, the SVM and KNN models consistently achieved the highest prediction accuracies across all three feature selection methods (Chi-Square, LASSO, and SFS), with accuracies of 94.31%, 95.79%, and 94.07%, respectively. In contrast, when SMOTE-ENN was not applied, XGBoost (72.83%), Random Forest (72.40%), and CatBoost (72.86%) yielded the highest accuracies across the three feature selection methods. Without SMOTE-ENN, CatBoost gives the most accuracy (72.86% among the three methods), but with SMOTE-ENN, SVM attains the maximum accuracy (95.79% within the same set of methods). These findings imply that the application of SMOTE-ENN greatly enhances prediction accuracy in comparison to Without-SMOTE-ENN. The careful tuning of hyperparameters is crucial to optimizing the performance of ML algorithms. All ML algorithms were initialized with a set of default parameters here. While some algorithms may not perform optimally with these default settings, their performance can be significantly improved by identifying and applying the best combination of hyperparameters (HP). To ensure optimal model performance, hyperparameter tuning was performed for all machine learning algorithms using a grid search strategy with 5-fold cross-validation. The hyperparameters (initial and optimal) that influenced the models' training and performance are presented in Table 5.

Using these hyperparameters, the predictive performance of the default and tuned classification algorithms was evaluated and compared with balancing datasets using performance metrics, resulting in high scores for each algorithm (Table 6).

**Table 3. Percentages of data distribution before and after applying SMOTE-ENN.**

| Methods | Before applying SMOTE-ENN | | After applying SMOTE-ENN | |
|---|---|---|---|---|
| | Underweight/Normal | Overweight/Obese | Underweight/Normal | Overweight/Obese |
| Filter | 67.2 | 32.8 | 37.6 | 62.4 |
| Embedded | | | 49.0 | 50.1 |
| Wrapper | | | 36.2 | 69.8 |

**Table 4. Accuracy (%) of balanced and imbalanced data for the default model.**

| | Chi-Square | | LASSO | | SFS | |
|---|---|---|---|---|---|---|
| Models | Balanced | Imbalanced | Balanced | Imbalanced | Balanced | Imbalanced |
| SVM | 94.31 | 71.42 | 95.79 | 72.22 | 92.12 | 71.42 |
| KNN | 93.98 | 70.01 | 95.13 | 70.10 | 94.07 | 70.44 |
| RF | 93.76 | 71.88 | 95.02 | 72.40 | 92.43 | 71.67 |
| XGBoost | 92.08 | 72.83 | 94.53 | 72.34 | 92.74 | 71.88 |
| CatBoost | 89.80 | 72.16 | 91.58 | 72.00 | 92.99 | 72.86 |
| MLP | 89.26 | 71.91 | 87.42 | 72.13 | 88.54 | 72.28 |
| LR | 83.95 | 71.57 | 79.98 | 71.57 | 84.71 | 71.45 |

**Table 5. Optimized hyperparameters after using the SMOTE-ENN technique.**

| Models | Initial Parameters | Optimized Parameters | | |
|---|---|---|---|---|
| | | Filter method | Embedded method | Wrapper method |
| SVM | 'C': [0.1, 1, 10, 100, 1000]<br>'gamma': [1, 0.1, 0.01, 0.001, 0.0001]<br>'kernel': ['rbf', 'linear'] | 'C': 10<br>'gamma': 1<br>'kernel': 'rbf' | 'C': 10<br>'gamma': 1<br>'kernel': 'rbf' | 'C': 10<br>'gamma': 0.1<br>'kernel': 'rbf' |
| LR | "penalty": ["l1","elasticnet"]<br>"solver": ["lbfgs","liblinear","sag","saga"]<br>"max_iter": [100,300,500,1000]<br>'C':np.linspace(0,1,30) | 'penalty': 'l1'<br>'solver': 'liblinear'<br>'max_iter': 100<br>'C': 0.172 | 'penalty': 'l1'<br>'solver': 'saga'<br>'max_iter': 100<br>'C': 0.172 | 'penalty': 'l2'<br>'solver': 'sag'<br>'max_iter': 100<br>'C': 0.172 |
| RF | 'n_estimators': [25, 50, 100, 150]<br>'max_features': ['sqrt', 'log2']<br>'min_samples_split': [2, 5, 10]<br>'min_samples_leaf': [1, 2, 4]<br>'bootstrap': [True, False]<br>'criterion': ['gini', 'entropy'] | 'n_estimators': 100<br>'max_features': 'sqrt'<br>'min_samples_split': 2<br>'min_samples_leaf': 1<br>'bootstrap': False<br>'criterion': 'entropy' | 'n_estimators': 100<br>'max_features': 'sqrt'<br>'min_samples_split': 2<br>'min_samples_leaf': 1<br>'bootstrap': False<br>'criterion': 'gini' | 'n_estimators': 100}<br>'max_features': 'sqrt'<br>'min_samples_split': 2<br>'min_samples_leaf': 1<br>'bootstrap': False<br>'criterion': 'gini' |
| KNN | "n_neighbors": [3, 4, 5, 8, 10]<br>'metric': ['euclidean', 'manhattan']<br>"weights": ["uniform","distance"]<br>"algorithm": ["auto","ball_tree","kd_tree","brute"]<br>"leaf_size": [10, 20, 30, 40, 50] | 'n_neighbors': 4<br>'metric': 'manhattan'<br>'weights': 'distance'<br>'algorithm': 'kd_tree'<br>'leaf_size': 20 | 'n_neighbors': 4<br>'metric': 'manhattan'<br>'weights': 'distance'<br>'algorithm': 'auto'<br>'leaf_size': 30 | 'n_neighbors': 4<br>'metric': 'euclidean'<br>'weights': 'distance'<br>'algorithm': 'kd_tree'<br>'leaf_size': 30 |
| XGBoost | 'learning_rate': [0.01, 0.1, 0.2]<br>'n_estimators': [100, 200, 300]<br>'max_depth': [3, 4, 5, 6]<br>'min_child_weight': [1, 2, 3]<br>'subsample': [0.6, 0.7, 0.8, 0.9]<br>'colsample_bytree': [0.6, 0.7, 0.8, 0.9]<br>'gamma': [0, 0.1, 0.2] | 'learning_rate': 0.2<br>'n_estimators': 300<br>'max_depth': 6<br>'min_child_weight': 1<br>'subsample': 0.9<br>'colsample_bytree': 0.7<br>'gamma': 0 | 'learning_rate': 0.2<br>'n_estimators': 300<br>'max_depth': 6<br>'min_child_weight': 1<br>'subsample': 0.8<br>'colsample_bytree': 0.9<br>'gamma': 0 | 'learning_rate': 0.2<br>'n_estimators': 300<br>'max_depth': 6<br>'min_child_weight': 1<br>'subsample': 0.9<br>'colsample_bytree': 0.8<br>'gamma': 0.1 |
| CatBoost | 'depth': [4, 6, 8]<br>'learning_rate': [0.01, 0.05, 0.1]<br>'iterations': [100, 200, 300]<br>'l2_leaf_reg': [1, 3, 5, 7, 9] | 'depth': 8<br>'learning_rate': 0.1<br>'iterations': 300<br>'l2_leaf_reg': 1 | 'depth': 8<br>'learning_rate': 0.1<br>'iterations': 300<br>'l2_leaf_reg': 3 | 'depth': 8<br>'learning_rate': 0.1<br>'iterations': 300<br>'l2_leaf_reg': 1 |
| MLP | 'hidden_layer_sizes': [(50,), (50,50), (100,), (100,50)]<br>'activation': ['tanh', 'relu','logistic']<br>'solver': ['sgd', 'adam']<br>'alpha': [0.0001,0.001,.01, 0.05]<br>'learning_rate':['constant','adaptive','invscaling'] | 'hidden_layer_sizes': (100, 50)<br>'activation': 'tanh'<br>'solver': 'adam'<br>'alpha': 0.0001<br>'learning_rate': 'constant' | 'hidden_layer_sizes': (100, 50)<br>'activation': 'relu'<br>'solver': 'adam'<br>'alpha': 0.0001<br>'learning_rate': 'constant' | 'hidden_layer_sizes': (100, 50)<br>'activation': 'tanh'<br>'solver': 'adam'<br>'alpha': 0.01<br>'learning_rate': 'constant' |

**Table 6. Accuracy (%) of the default and HP-tuned data for the balancing dataset.**

| Model | Filter | | Embedded | | Wrapper | |
|---|---|---|---|---|---|---|
| | Default | HP tuned | Default | HP tuned | Default | HP tuned |
| SVM | 87.04 | **94.31** | 82.28 | **95.79** | 85.98 | **92.12** |
| LR | 83.79 | **83.95** | 79.87 | **79.98** | 84.6 | **84.71** |
| RF | 92.84 | **93.76** | 94.2 | **95.02** | 91.92 | **92.43** |
| KNN | 91.59 | **93.98** | 91.36 | **95.13** | 90.9 | **94.07** |
| XGB | 91.54 | **92.08** | 93.98 | **94.53** | 91.66 | **92.74** |
| CB | 88.34 | **89.8** | 89.33 | **91.58** | 91.2 | **92.99** |
| MLP | 87.36 | **89.26** | 84.25 | **87.42** | 87.26 | **88.54** |

Thereafter, the classification models were evaluated using a range of performance metrics, including accuracy, precision, recall, F1-score, Cohen's kappa, and specificity, and the results were compared for each of the three feature selection techniques. In the case of the filter method, SVM, KNN, and RF demonstrate superior performance, achieving the highest values for each evaluated performance metric (Fig 3).

Among these, SVM stands out by producing the best results in several categories, including accuracy (94.31%), recall (97.83%), F1-score (95.55%), and kappa (87.67%). The prediction performance of RF was higher than that of other models in terms of specificity (89.47%) and precision (93.82%). Moreover, the RF algorithm consistently outperformed the other models overall. Among the evaluated models, SVM achieved the highest AUC at 93.14%, followed closely by KNN at 92.94% and RF at 92.91% (S1 Fig).

For the embedded method, SVM, and KNN demonstrate superior performance, achieving the highest values for each performance metric evaluated in Fig 4. Among these, SVM stands out by producing the best results in several categories, including accuracy (95.79%), precision (97.32%), F1-score (95.77%), kappa (91.58%), and specificity (97.34%). The

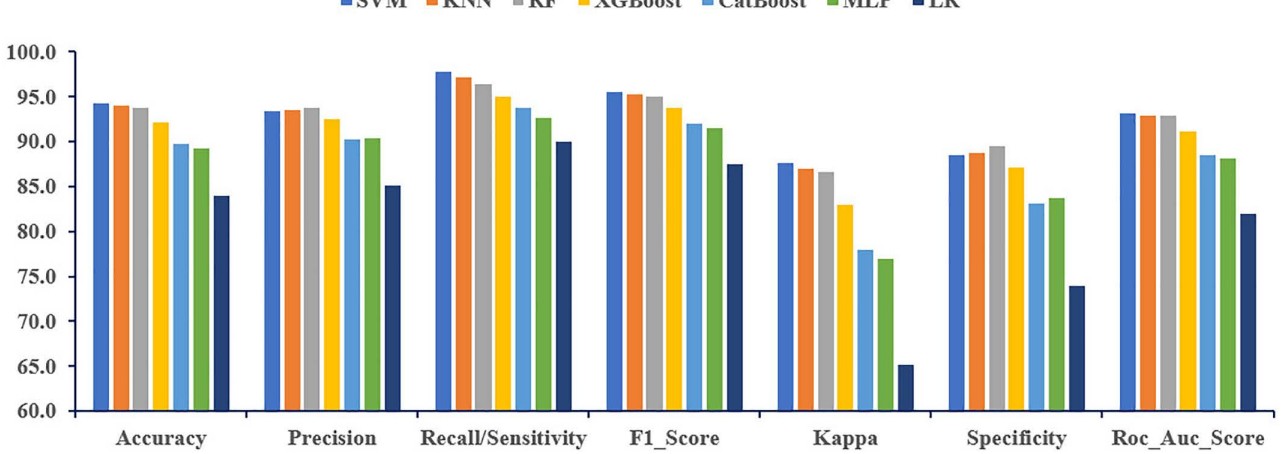

**Fig 3. Performance comparison of ML classifiers for the Chi-square (Filter).**

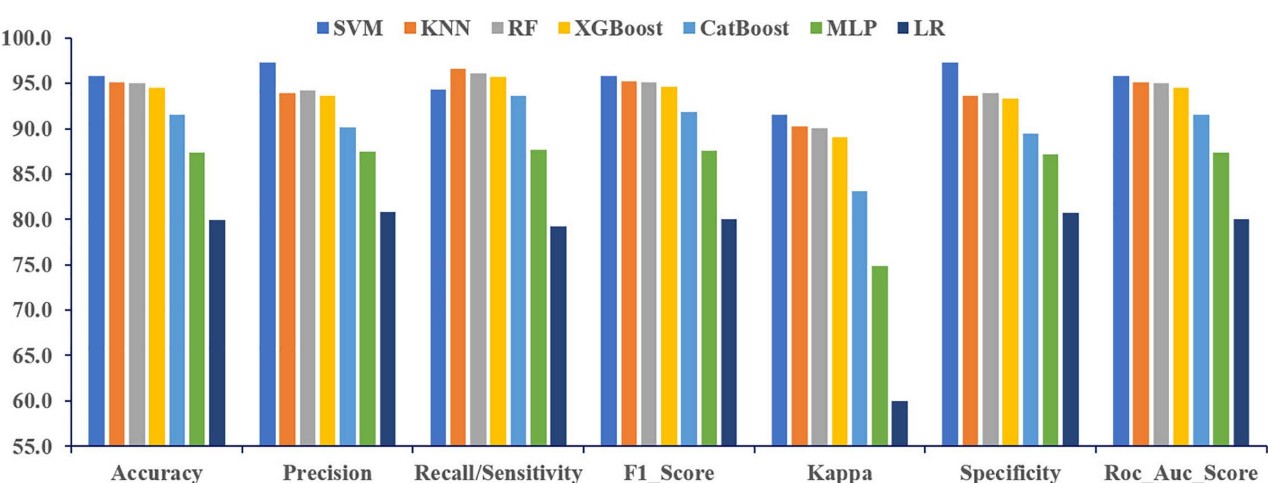

**Fig 4. Performance comparison of ML classifiers for the LASSO (Embedded).**

prediction performance of KNN was higher than that of other models in terms of corresponding recall (96.65%). Moreover, the RF algorithm consistently outperformed the other models overall. The AUC outperformed for the SVM algorithm, achieving 95.81% (S1 Fig). Among the other algorithms, KNN achieved 95.11%, RF achieved 95.01%, and XGBoost reached 94.52%, each demonstrating strong performance in their respective evaluations accordingly.

In the wrapper method, both KNN and XGBoost exhibited excellent performance, achieving the top scores across all performance metrics (Fig 5). KNN particularly excelled, delivering the highest results in categories such as accuracy (94.07%), recall (98.83%), F1-score (95.64%), and kappa (86.39%). On the other hand, XGBoost outperformed other models with 92.69% precision and 85.33% specificity. Additionally, CatBoost consistently outperformed the other models. The highest AUC was observed for the KNN algorithm, achieving 91.86% (S1 Fig). Among the remaining models, CatBoost achieved an AUC of 91.11%, while XGBoost recorded 90.96%, each demonstrating strong performance in their respective evaluations

The embedded feature selection technique yielded the best overall performance for the SVM, KNN, and RF algorithms, as evidenced by superior results across various metrics, including accuracy, precision, Cohen's kappa, specificity, and AUC. To evaluate the effectiveness of the proposed algorithm, its predictive accuracy was compared with that of several other machine learning models (Fig 4). The SVM model consistently demonstrated superior predictive performance across all comparisons. The accuracy values (95.79%) for the SVM model were notably high, indicating strong agreement with the data. For the dataset from BDHS 2017–2018, the SVM model outperformed all other models in performance metrics, except for recall, where it still showed competitive results relative to LR, RF, KNN, XGBoost, CatBoost, and MLP. The proposed SVM model stood out with excellent results, achieving 95.79% accuracy, 97.32% precision, 95.77% F1-score, 91.58% kappa, 97.34% specificity, and 95.81% AUC, highlighting its strong predictive power over the other evaluated models.

Another proposed KNN model was compared in terms of predictive accuracy against seven other ML algorithms. With 95.13% accuracy, the KNN model ranked second overall, while achieving the highest recall value (96.65%), reflecting a strong alignment with the data. The KNN model outperformed all models except SVM in key performance metrics, including accuracy (95.13%), F1-score (95.26%), kappa (90.26%), and AUC (95.11%). It demonstrated competitive performance compared to models such as LR, RF, XGBoost, CatBoost, and MLP. Furthermore, the next proposed RF model was also assessed and compared. The RF model outperformed all models except SVM and KNN in key performance metrics, including accuracy (95.02%), precision (94.17%), recall (96.11%), F1-score (95.13%), kappa (90.04%), specificity (93.91%), and AUC (95.01%). With an accuracy of 95.02%, the RF model ranked third in terms of accuracy, F1-score, kappa, and AUC. It achieved the second-highest performance in terms of precision, recall, and specificity. For comparison, logistic regression was included as a classical statistical model. While LR achieved an accuracy of up to 84.71%

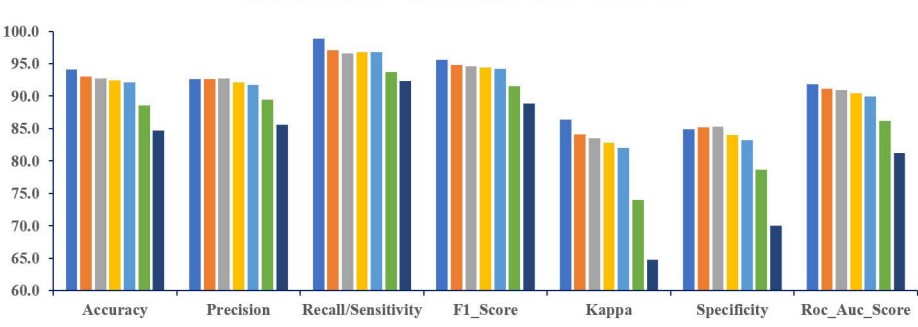

**Fig 5. Performance comparison of ML classifiers for the SFS (Wrapper).**

(after hyperparameter tuning), the best-performing machine learning models, including SVM (95.79% accuracy) and KNN (95.13%), substantially outperformed LR (Table 6). Therefore, it can be concluded that, given the consistent results and selected features, the embedded method was identified as the best feature selector. The best ML models are SVM, followed by KNN and RF, as they demonstrate robustness and suitability.

Chi-square, SFS, and Lasso (L1 regularization) were used from the Filter, Wrapper, and Embedded methods, respectively, to identify the most relevant subsets of features. For the Filter method, 19 features were selected based on their statistical significance, as determined by the P-value and high Chi-square value criteria (S2 Table). Features with a P-value less than 0.05 (P < 0.05) were selected, indicating that these features are statistically significant and have a strong association with the overweight/obesity status. These selected features demonstrated high prediction relevance in the model and contributed to improved classification performance. In the case of the Embedded method, LASSO feature selection with a regularization parameter set to alpha = 0.01 was employed, resulting in the selection of 12 features with non-zero coefficients. These features, which have non-zero Lasso coefficients, are identified as significant and retained in the model due to their contribution to the predictive performance. Additionally, the wrapper method was employed for feature selection, applying the sequential feature selector (SFS) technique. Forward selection was specifically employed, wherein features were iteratively added according to their contribution to improving the model's classification accuracy. Using the best-performing model (KNN), this process resulted in the selection of 23 key features.

LASSO selected a more parsimonious feature set (12 features) while achieving the highest model performance (SVM accuracy: 95.79%), demonstrating its efficiency in both dimensionality reduction and predictive accuracy. Most features (10/12) corroborate previously identified risk factors, supporting the validity of the method (Table 7).

Therefore, the embedded method selected a smaller subset of features (12 features), all of which were also chosen by the other feature selection methods (Fig 6). The filter method identified 7 unique features not selected by the embedded method. In comparison, the wrapper method selected 11 unique features beyond those chosen by the embedded method, as well as 6 additional features not captured by the filter method.

## 3.2 Feature importance

The permutation importance values derived from the best-performing model—Support Vector Machine (SVM)— reflect the relative influence of each feature on prediction accuracy (Fig 7). The feature V190 (wealth index combined) holds the highest importance of 0.239, indicating its dominant role in shaping the model's decisions. Nutrition education programs and community-based lifestyle interventions should target wealthier women to encourage balanced diets and active lifestyles. This is followed by 0.185 for V013 (age in 5-year groups) and 0.174 for V024 (division), which also contribute significantly to the model's predictive capability. Moderately important features include 0.138 for V701 (husband/partner's education), 0.124 for V159 (frequency of watching television), and 0.098 for V201 (total children ever born), whereas 0.097 for V704 (husband/partner's occupation), 0.094 for V106 (respondent's highest educational level), and 0.027 for V025 (type of place of residence) exhibit relatively lower influence. Public health campaigns could prioritize middle-aged women for obesity screening and age-appropriate physical activity programs. The least impactful features in this study are V122 (household has a refrigerator) (0.019), V404 (currently breastfeeding) (0.018), and V161 (type of cooking fuel) (0.011), indicating their minimal effect on classification performance.

Likewise, the SHAP-based feature importance scores from the SVM model indicate that the wealth index (V190) is the most significant predictor of overweight/obesity (0.1584), suggesting that economic status has a strong influence on overweight/obesity. Age group (V013, 0.0972) and frequency of watching television (V159, 0.0766) also play major roles, linking obesity to age-related metabolic changes and sedentary behavior. Division (V024, 0.0711), husband's education (V701, 0.0626), and respondent's education (V106, 0.0461) have a moderate impact on obesity prediction, suggesting that education level and location affect lifestyle choices. Features such as occupation (V704, 0.0440),

**Table 7. Existing studies on overweight and obesity among ever-married women.**

| Feature Name | Selected by FS Approach | Existing Studies |
|---|---|---|
| Age in 5 years | Filter, Wrapper, Embedded | [8,55–70] |
| Respondent's highest educational level | Filter, Wrapper, Embedded | [ 8,55, 15,57–70] |
| Division | Filter, Wrapper, Embedded | [ 8,57,15,60–70] |
| Type of place of residence | Filter, Wrapper, Embedded | [ 8,55,56,15, 59–70] |
| Wealth index combined | Filter, Wrapper, Embedded | [ 8,55,57,58,15, 61–70] |
| Frequency of watching television | Filter, Wrapper, Embedded | [57,59–61,8,67–69] |
| Husband/partner's education level | Filter, Wrapper, Embedded | [59,61,8,63,64,67–69] |
| Total children ever born | Filter, Wrapper, Embedded | [55,59,61,63,64,67,69,15] |
| Current contraceptive method | Wrapper | [57,60,62,8,69] |
| Husband/partner's occupation | Filter, Wrapper, Embedded | [69,70] |
| Currently breastfeeding | Filter, Wrapper, Embedded | [57,67] |
| Household has: refrigerator | Filter, Wrapper, Embedded | |
| Type of cooking fuel | Filter, Wrapper, Embedded | |
| Wealth index for urban/rural | Filter, Wrapper | |
| Household has: television | Filter, Wrapper | |
| Household has: motorcycle/scooter | Filter | |
| Type of toilet facility | Filter, Wrapper | |
| Household has: electricity | Filter, Wrapper | |
| Household has: car/truck | Filter | |
| Currently pregnant | Filter, Wrapper | |
| Sex of household head | Wrapper | |
| Household has: bicycle | Wrapper | |
| Respondent's occupation | Wrapper | |
| Source of drinking water | Wrapper | |
| Person who usually decides on the respondent's health care | Wrapper | |

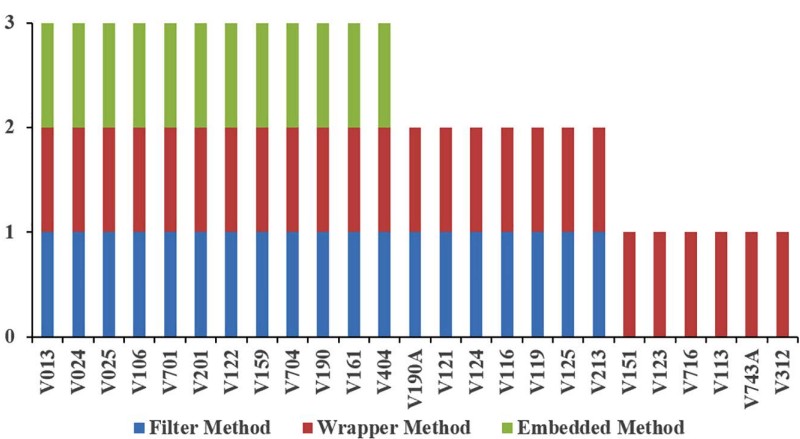

**Fig 6. Number of selected features using three feature selection methods.**

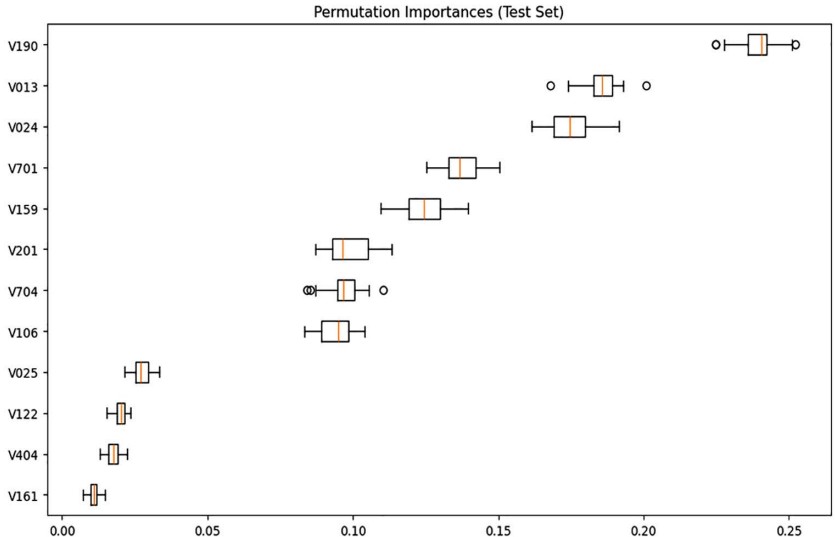

**Fig 7. Permutation importance of features applied to the proposed SVM model.**

number of children (V201, 0.0389), breastfeeding status (V404, 0.0305), urban/rural residence (V025, 0.0288), and refrigerator ownership (V122, 0.0258) have lower predictive influence but still contribute to lifestyle-related obesity factors. The type of cooking fuel (V161, 0.0139) has the least impact, showing minimal relevance to obesity prediction in the model (Fig 8).

The SHAP violin plot provides a distributional view of how each predictor influences the model's output across all individuals. For key variables such as wealth index (V190), age group (V013), and television watching frequency (V159), higher feature values (shown in pink) are predominantly associated with positive SHAP values, indicating an increased

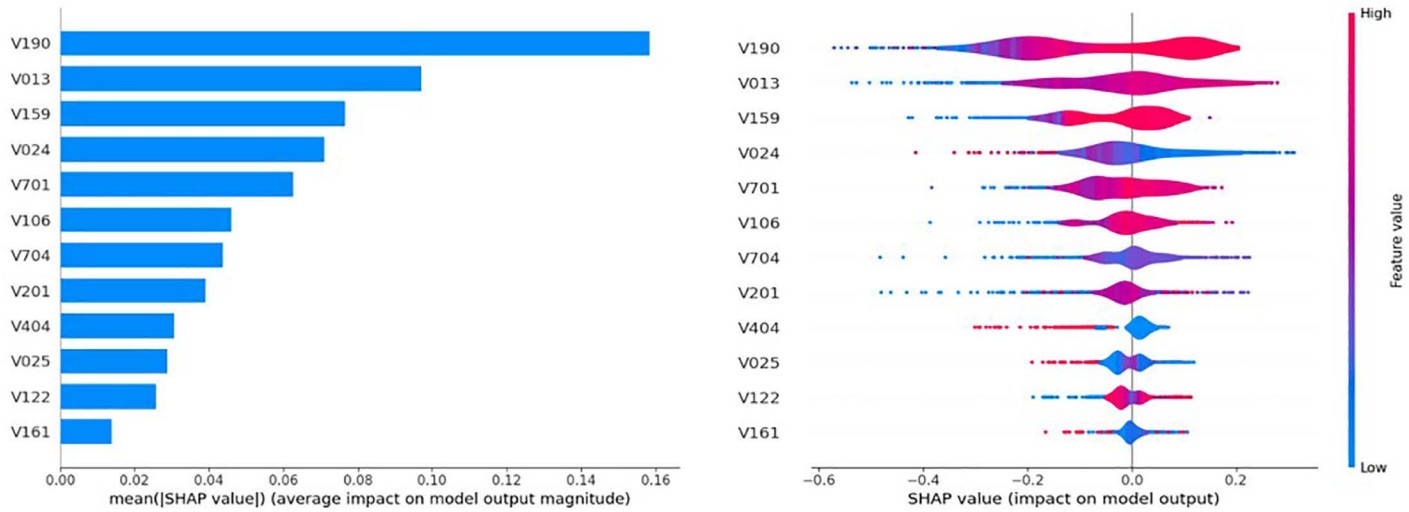

**Fig 8. SHAP-based global feature importance and feature-wise contribution patterns for the SVM model.** (a) Global feature importance ranked by mean absolute SHAP values for the SVM model.; (b) SHAP violin plot illustrating the feature-wise contribution patterns and their influence on the model output.

predicted risk of obesity. Lower feature values (blue) shift SHAP values toward negative values, suggesting a reduced risk. For educational variables (V701, V106), higher education levels tend to increase predictions toward the obese class, while lower values reduce the predicted probability, although with greater variability than the primary predictors. Geographic and household characteristics, such as division (V024), urban/rural status (V025), and refrigerator ownership (V122), exhibit more dispersed SHAP distributions, indicating that their effects vary substantially among individuals. Some predictors, specifically cooking fuel type (V161) and breastfeeding status (V404), display SHAP values that are tightly concentrated around zero, indicating a weak influence on the model's decisions. Meanwhile, occupation (V704) and the number of children (V201) exhibit broader distributions, suggesting heterogeneous effects that depend on individual circumstances.

### 3.3 Discussion

Numerous studies have been reviewed and found a higher prevalence of overweight/obesity among ever-married Bangladeshi women using traditional methods. These findings highlight an increased risk of obesity with increasing age [8,15,55–70], especially among women aged 30–50, who exhibit the highest prevalence rates [8,55,57,69]. In contrast, the lowest rates are observed among those aged 15–24 [55,57]. Trends in overweight and obesity have shown variations across geographical divisions, with studies reporting prevalence across Barishal, Chittagong, Dhaka, Rajshahi, and Sylhet [8,57,15,60–70]. Many studies identified that rural women tended to have a higher incidence of underweight, while urban women consistently showed a higher incidence of overweight or obesity [8,55,56,15, 60–70]. The educational levels of both the respondent and her husband significantly influenced overweight and obesity rates. Women with secondary or higher education, and those whose husbands were also educated, exhibited a greater likelihood of being overweight or obese [8,15, 57–71]. Furthermore, women's BMI was notably higher when their husbands were educated. Based on a husband's occupation, the prevalence of normal weight is lowest in higher education and highest in overweight and obesity [69,70]. The number of children ever born [55,59,61,63,64,67,69,15] and contraceptive use [57,60,62,8,69] were strongly associated with overweight or obesity. Additionally, the likelihood of being underweight was greater among women who were not engaged in breastfeeding [57,67]. Frequent television viewing (≥ once a week) has been associated with an increased prevalence of obesity, while those who watched less frequently had lower odds of obesity [57,59–61,8,67–69]. Interestingly, women with minimal TV exposure (<once/week) were more likely to be obese than those who never watched [64,8,68]. Women's overweight or obesity status was strongly influenced by their household wealth status. As household wealth increased, so did the prevalence of overweight and obesity. In contrast, underweight was most common among women from the poorest households [8,55,57,58,15,61–70].

Most of the key features, when compared to other studies, were selected using the embedded method rather than the filter or wrapper methods (Table 7). The embedded method selected 12 features, 10 of which have been identified in other studies as having a significant impact on overweight/obesity. In contrast, the filter and wrapper methods selected 19 and 23 features, respectively, with 10 and 11 features being significant, as indicated by prior research. Notably, the wrapper method identified "Current contraceptive method" as a highly significant feature in some studies, which was not selected by the embedded method despite its importance. This suggests that although the embedded method selects fewer features, previous studies have recognized nearly all as impactful, making it more efficient than both the filter and wrapper methods.

Consequently, based on the prior studies, the permutation analysis, and the SHAP-based analysis; the top six features—wealth index combined, age in 5-year group, division, respondent's education, husband/partner's education, and frequency of watching television—are the most significant predictors of overweight/obesity, as identified through the embedded method. The identified predictors offer actionable insights for public health planning in Bangladesh: interventions should focus on wealthier, middle-aged, urban women, leverage education and health literacy campaigns, and target lifestyle modifications, including reducing sedentary behaviors and promoting active living. For example, our analysis

identified "Wealth Index", "Age" and "Division" as top predictors. Policymakers can utilize these model predictions to inform them of their prioritization of resource allocation. Public health departments can allocate more resources to facilities in high-risk divisions (e.g., Chittagong and urban centers), rather than applying a uniform strategy across all regions. In addition, the strong influence of "Frequency of watching television" highlights a specific channel for intervention. Public health campaigns promoting physical activity could be strategically placed during peak television viewing times to target the most sedentary and at-risk demographic.

## 4.  Summary and conclusion

Overweight or obesity are recognized as significant epidemiological concerns, as they are associated with a range of chronic health conditions, including hypertension, type 2 diabetes mellitus, osteoarthritis, stroke, certain types of cancer, and even premature mortality. Globally, the prevalence of overweight and obesity has risen substantially, and approximately 50% of the global population will be classified as overweight by 2030. Overweight/obesity has become a significant health concern even for developing countries like Bangladesh, and women are relatively more at risk of being obese than men here, thereby fostering a strong research interest in this area. In Bangladesh, women who have ever been married often experience distinct social and cultural pressures, changes in lifestyle after marriage, and reproductive health challenges that can influence weight gain. Economic constraints, limited access to health education, and traditional gender roles may further exacerbate the risk of being overweight/obesity in this group. By targeting this population, our study provides context-specific insights that are critical for designing effective, culturally sensitive interventions aimed at improving women's health outcomes in Bangladesh.

Many existing studies examined the incidence and contributing factors of overweight and obesity among Bangladeshi adults through traditional statistical techniques with lesser operational performances [8–12]. Since machine learning enables the identification of complex patterns within large datasets, it can be effectively utilized to accurately identify key predictors associated with multifaceted conditions such as overweight and obesity. Therefore, this study presents a comprehensive analysis aimed at identifying the threatening factors and predicting the likelihood of overweight/obesity using various ML classifiers among ever-married Bangladeshi women.

In this study, the data used for model development were sourced from the nationally representative Bangladesh Demographic and Health Survey (BDHS) 2017–2018. Furthermore, several ML models, such as SVM, KNN, RF, XGBoost, CatBoost, MLP, and LR, were tested to develop a smart model for recognizing the risk factors of being obese or overweight, which will serve as a decision-support tool for specialists in the field. The proposed model for obesity prediction aims to achieve improved classification accuracy. This approach integrates three feature selection techniques—Chi-Square, LASSO, and SFS—with preprocessing and classification methods, to deliver promising diagnostic results. Later, the importance of the features was measured through the use of Permutation and SHAP analysis. The LASSO (embedded) method selects fewer features, with most being influential, making it more efficient compared to the filter and wrapper methods.

The Synthetic Minority Oversampling Technique and Edited Nearest Neighbor (SMOTE-ENN) data balancing approach was applied to the imbalanced dataset, and the performance of classification was compared with that of the entire dataset. The comparison between machine learning models with and without SMOTE-ENN reveals that classifiers trained on balanced datasets consistently outperform those trained on imbalanced datasets. Furthermore, 5-fold cross-validation was used, and the hyperparameters of the tested models were optimized. Balancing the dataset led to improved performance across all models. The findings confirmed that SVM with LASSO selection provided the optimal balance of metrics (Accuracy: 95.79%, AUC: 95.81%), outperforming ensemble methods like Random Forest and XGBoost.

The study also revealed that the key risk factors in predicting obesity include age, geographic division, place of residence, education levels of both the respondent and their partner, the partner's occupation, number of children ever born, frequency of television watching, and wealth status. From the feature importance of permutation analysis and SHAP analysis, it is evident that key risk factors, including wealth status, age, and frequency of watching television, have a significant

influence on overweight and obesity, suggesting that economic status and age have a significant impact on being over-weight/obese. Likewise, geographic division and the education levels of both the respondent and their partner moderately impact obesity prediction, suggesting that education level and location affect lifestyle choices.

Therefore, it can be concluded that the LASSO (Embedded) feature selection method effectively identified the most significant features, while the SVM algorithm proved to be the best classification model. These selected features not only reduced the computational burden but also highlighted the most relevant information. While logistic regression captured primarily linear associations between features and overweight/obesity outcomes, permutation and SHAP analyses indicated that SVM effectively modeled nonlinear interactions, such as those between wealth index combined, age in 5-year group, division, respondent's education, husband/partner's education, and frequency of watching television, thereby providing superior predictive performance. The study also concludes that machine learning can be a powerful tool in public health, enabling timely predictions for individuals at risk of being overweight/obesity, which could assist decision-makers in taking immediate preventive actions.

## Supporting information

**S1 Table. Bivariate analysis of factors associated with Underweight/Normal and Overweight/Obese among ever-married women in Bangladesh.**
(DOCX)

**S2 Table. Feature selection by Filter, Wrapper, and Embedded methods.**
(DOCX)

**S1 Fig. AUC comparison of ML classifiers with Chi-square, LASSO and SFS.**
(TIF)

**S1 File. Processed data.**
(CSV)

## Author contributions

**Conceptualization:** Suman Biswas, Md. Mahamudul Islam, Nusrat Islam.

**Data curation:** Suman Biswas, Md. Mahamudul Islam, Nusrat Islam, Md. Abdur Rahim Mia.

**Formal analysis:** Suman Biswas, Md. Mahamudul Islam, Nusrat Islam, Md. Abdur Rahim Mia.

**Funding acquisition:** Suman Biswas.

**Methodology:** Suman Biswas, Md. Mahamudul Islam, Nusrat Islam, Md. Abdur Rahim Mia.

**Software:** Suman Biswas, Md. Mahamudul Islam, Nusrat Islam, Md. Abdur Rahim Mia.

**Supervision:** Suman Biswas.

**Validation:** Suman Biswas, Md. Mahamudul Islam, Nusrat Islam, Md. Abdur Rahim Mia.

**Visualization:** Suman Biswas, Md. Mahamudul Islam, Nusrat Islam, Md. Abdur Rahim Mia.

**Writing – original draft:** Suman Biswas, Md. Mahamudul Islam, Nusrat Islam, Md. Abdur Rahim Mia.

**Writing – review & editing:** Suman Biswas.

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
