## [Decision Letter · Decision Letter 0]

26 Oct 2025

Dear Dr. Biswas,

Thank you for submitting your manuscript to PLOS ONE. After careful consideration, we feel that it has merit but does not fully meet PLOS ONE’s publication criteria as it currently stands. Therefore, we invite you to submit a revised version of the manuscript that addresses the points raised during the review process.

We look forward to receiving your revised manuscript.

Kind regards,

Guanghui Liu

Academic Editor

PLOS ONE

4. We are unable to open your Supporting Information file [Processed_data_BDHS_2017-2018.sav]. Please kindly revise as necessary and re-upload.

Reviewers' comments:

Reviewer's Responses to Questions

**Comments to the Author**

1. Is the manuscript technically sound, and do the data support the conclusions?

Reviewer #1: Yes

Reviewer #2: Yes

Reviewer #3: Yes

Reviewer #4: Yes

2. Has the statistical analysis been performed appropriately and rigorously?

Reviewer #1: Yes

Reviewer #2: Yes

Reviewer #3: Yes

Reviewer #4: Yes

3. Have the authors made all data underlying the findings in their manuscript fully available?

Reviewer #1: Yes

Reviewer #2: Yes

Reviewer #3: Yes

Reviewer #4: Yes

4. Is the manuscript presented in an intelligible fashion and written in standard English?

Reviewer #1: Yes

Reviewer #2: No

Reviewer #3: Yes

Reviewer #4: Yes

Reviewer #1: Biswas et al. presented a study identifying risk factors of Bangladeshi women overweight and obesity. They applied and compared different machine learning and feature selection methods, finding that SVM combining with the LASSO technique achieves the best results. The study also suggests that wealth status, age and frequency of watching TV have strong influences of overweight and obesity. It is an interesting topic but I think the authors need to address the following issues:

Major:

1. Is there any reasons the authors need to use the codes instead of the names of the features? For example, 'age' for feature 'V013'. It is not easy for readers to follow the results part, especially for understanding figure 9 and 10.

2. It is unclear what parameters were used for SMOTE-ENN and what is the number for underweight/normal and overweight/obese data before and after addressing the class imbalance issue. It would be better if the authors can provide such numbers to make the manuscript transparent and reproducible.

3. Section 3.1 is not very easy to follow, especially from line 340 to 364. The title of this section is 'Results of ML classifier' but these lines to me are like comparing feature selection methods instead of ML classifiers. I think it would be clearer if it can be rephrased like for method xxx, xxx feature selection method gives the best results with accuracy xxx, recall xxx, etc., since the main purpose of the manuscript is to find the best model and interpret the results instead of comparing different feature selecting methods.

4. The interpretation of Figure 10b is not clear. The waterfall plot explains the prediction for single row instead of the whole data (this is why there is gray text before the feature codes showing the value of each feature for this specific data point). The figure 10 legend is a kind of misleading that both panel a and b are about the whole dateset.

minor:

1. In line 45-47, the authors compared the obesity percentage of adults in Bangladesh in 1990 and 2022. Since the manuscript is about ever-married Bangladeshi women, is there any statistic (i.e. obesity percentage) of ever-married Bangladeshi women or all Bangladeshi women in 1990 and 2022?

2. In the introduction previous study part, the authors mentioned L2 penalty was used for feature selection and algorithm comparison (line 69). Is there any specific reason the authors used L1 (LASSO) penalty in the study?

3. In line 516-517 '...suggesting economic status and age strongly influences weight gain.' In this study, the authors only focus on overweight/obesity (static state) instead of weight gain (dynamic process).

4. In line 507-511, since the data is imbalanced, it would be better to report precision/recall/F1-score instead accuracy when comparing the performance of different models.

Reviewer #2: In this paper, the authors have systematically compared the performance of a variety of machine learning models in predicting the overweight/underweight among women in Bangladeshi. The background information is great, and the comparison is valid. I appreciate that the authors have shared the code in a public repository which is very helpful for reproducible research and potentially a good educational resource.

However, the novelty of this study is lacking. The author neither proposed new approach nor provided an improvement of the current methods. The conclusion and findings are consistent with previous studies, and no new aspects were concluded in this study. Most importantly, the author did not include a comparison between the machine learning model and the traditional methods such as logistic regression. A discussion on the advantages and disadvantages of the machine learning model compared to the classic methods in predicting the overweight/underweight is important. For example, in overweight/underweight problems, identifying the more relevant predictors might be more important than the prediction accuracy itself. In that case, the traditional method might be better in terms of explainability.

My other comments are:

I suggest removing Figure 3 as the information (the number of underweight/overweight women) can be easily conveyed in the main text. This will save room for figures to illustrate more complex results. The information contained in Figure 3 does not contain relevant information. Why only women aged 15-49 were included? What about women who are 50+?

As hyperparameter tuning dramatically affects the model performance. How did the author make sure that the tuning parameters are optimized for each specific ML algorithm? The poor performance could be due to the suboptimal hyperparameters.

The author should proofread the manuscript. In line 322, the presented accuracy for the machine learning models does not match the result in Table 3. The author should make sure the reported number is consistent between tables and the main context. Two SVMs are mentioned in line 322, and I believe one of these two is a typo. Same problems occurred in line 325.

In line 298 “… on the on the respondent’s…”

I’m confused with the data presented in Figure 7. What is the difference between Figure 7 and Figure 5? In addition, some labels at x-axis are missing in Figure 7.

The author should clarify the results and their connections presented in Figure 10 (a) and 10 (b). For example, in figure 10 (a) the SHAP-based feature importance scores from the SVM model highlight that wealth index (V190) is the most significant predictor of obesity, however in figure 10 (b) the contribution from V190 is the lowest 0.01 compared to other features.

Reviewer #3: This study applied multiple machine learning algorithms—including SVM, KNN, RF, XGBoost, CatBoost, MLP, and Logistic Regression—to predict overweight and obesity among ever-married Bangladeshi women using nationally representative BDHS 2017–2018 data. Although the proposed methodology effectively identified key risk factors and achieved high predictive accuracy, the article needs improvements in the following aspects:

Introduction

1.Transitions between global and national-level data discussions and the introduction of machine learning techniques are abrupt.

Recommendation: Include a brief transitional sentence or phrase to smoothly move from global trends to national data.

2. The literature review primarily lists prior studies and methods without critically evaluating their strengths, weaknesses, or gaps.

Recommendation: Provide critical evaluation, highlighting limitations of previous methods (e.g., insufficient accuracy, inadequate handling of data imbalance), to justify your study clearly.

3. Certain expressions and word choices are informal or less scholarly, e.g.: "So, overweight/obesity is an alarming issue…", inconsistent tense ("were" should be "are").

Recommendation:

Adopt a more formal academic tone. For example: Replace "So" with "Therefore" or "Thus." Use "significant public health concern" instead of "alarming issue."

Consistently use "among" instead of "amidst." Correct tense consistency ("objectives are").

4. Figure references (e.g., "Fig 1") are presented casually and not formally.

Recommendation: Use a more formal figure referencing format, e.g., "Figure 1 illustrates…".

5. The stated objectives ("to select crucial factors and the importance of explaining...") are somewhat vague or ambiguous.

Recommendation: Clarify and rewrite objectives for precision, such as: "To identify key predictors and assess their contribution to explaining overweight and obesity among ever-married Bangladeshi women.

Materials & Methods

6. Since obesity and overweight are presented separately in Figure 1, please maintain consistency here in Table 1 as well.

7. The current dataset description lacks important contextual details such as the target variable (overweight/obesity) clearly stated.

Information about ethical considerations or permissions to use the dataset is not provided explicitly.

Recommendations:

Clearly specify the target variable early in the dataset description.

Briefly mention ethical approval or publicly available status explicitly.

8. Vague phrases are used, such as “necessary corrections were applied.” The type of corrections applied (e.g., imputation method) is not described.

The term "transformation" is ambiguous. It’s unclear what specific transformations were performed on the dataset.

Recommendations:

Clearly state how missing values and outliers were handled (e.g., removal, mean/median imputation, etc.).

Explicitly describe what "transformation" involved (e.g., categorical encoding, normalization, standardization).

9. The need for using SMOTE-ENN could be further strengthened with statistics on class imbalance.

Recommendations: Include specific imbalance statistics (e.g., percentages of majority vs. minority classes before and after applying SMOTE-ENN).

10. Although cross-validation is mentioned, specific details about which hyperparameters were tuned for each model are absent.

Recommendations: Clarify briefly which hyperparameters were optimized for each algorithm or clearly state that these details are provided in supplementary materials if available.

11.  The Methods section should provide sufficient procedural details to allow reproducibility by others. However, this manuscript lacks detailed descriptions of experimental procedures, especially in sections 2.7 through 2.11, where excessive general explanations and definitions of commonly understood terms and models are presented. Such textbook-style information is unnecessary in the main text. If the authors consider these explanations valuable, they should relocate this portion to supplementary materials to maintain clarity and conciseness in the primary manuscript.

Results and discussion

12. Some subsections are overly detailed with repetitive statements, especially regarding accuracy and performance metrics.

Results and discussion are integrated without clear delineation, making the narrative difficult to follow at times.

Recommendations:

Clearly separate the presentation of results from their interpretation.

Consider creating clear subsections (e.g., "Model Results," "Feature Importance," and then "Discussion").

Summarize repetitive numerical results into concise tables or figures, rather than lengthy paragraphs.

13. The results of bivariate analysis are only briefly mentioned without much elaboration or interpretation.

The significance of factors mentioned (age, wealth, education, etc.) could benefit from clearer interpretation.

Recommendations:

Include brief interpretations of why these factors might significantly associate with overweight/obesity, providing context from existing literature or theoretical justifications.

14. Results are primarily presented without sufficient integration into existing knowledge or practical implications. For instance, many references are cited, but their connections to your findings are briefly addressed.

Recommendations:

Strengthen the discussion by clearly interpreting how your ML-based findings confirm, expand upon, or differ from traditional analyses previously conducted in Bangladesh.

Discuss explicitly the practical implications of your key findings (wealth, age, education) for policy or public health programs in Bangladesh.

15. Figures are cited but not clearly described within the text, leaving readers to infer their significance.

Recommendations:

Add concise descriptive captions and explicitly refer to what readers should learn from each figure, ensuring the narrative clearly aligns with visual representations.

Summary and conclusion

16. The summary repeats certain points unnecessarily (e.g., statements about the superiority of ML methods and feature selection effectiveness appear repeatedly). Some sentences are lengthy and slightly unclear, diminishing readability.

Recommendations:

Combine repetitive statements into concise sentences.

Break down complex sentences into shorter ones for improved readability.

17. While global and Bangladesh-specific obesity statistics are mentioned, the unique significance of your study (why ever-married Bangladeshi women specifically) needs clearer reiteration here.

Recommendations:

Briefly restate why ever-married Bangladeshi women were chosen, highlighting unique vulnerabilities or contextual factors (social, economic, cultural) that underscore the importance of your findings.

18. Methodological steps (data balancing, feature selection, hyperparameter tuning) are described, but their novelty or distinctive advantage is not explicitly clarified.

Recommendations:

Explicitly summarize how your methodological approach differs or improves upon previous traditional analyses (e.g., improved accuracy, better handling of class imbalance, reduced computational complexity).

19. Statements regarding LASSO (embedded) feature selection being “efficient” are repeated multiple times without sufficient quantitative justification.

Recommendations:

Summarize briefly how and why the embedded (LASSO) method was superior (e.g., fewer features selected, higher predictive accuracy, stronger agreement with previous studies).

Clearly articulate how identified risk factors directly translate into actionable insights or interventions.

20. The performance of various models is extensively described, leading to redundancy.

Practical implications of model results for public health interventions are underemphasized.

Recommendations:

Summarize model results concisely, highlighting only the most impactful models (e.g., top 3) and their key metrics.

Explicitly state how your predictive model can practically inform policymakers or healthcare practitioners (e.g., targeted interventions, resource allocation, public health campaigns).

21. The interpretation of feature importance (age, wealth, education, etc.) and their link to obesity is briefly stated but lacks deeper context or policy recommendations.

Recommendations:

Briefly explain why these identified features matter practically in Bangladesh (e.g., implications for educational campaigns, socioeconomic interventions, or lifestyle-related public health strategies).

Reviewer #4: This manuscript presents a robust and timely application of multiple machine learning models to predict overweight and obesity among ever-married women in Bangladesh, leveraging recent nationally representative data. The authors systematically apply feature selection, data balancing, and model interpretation techniques to enhance both predictive performance and interpretability, yielding impressive accuracy metrics—particularly for the SVM model with embedded feature selection (LASSO). The study fills an important gap in the literature by advancing from traditional regression approaches to modern AI techniques in a public health context relevant to LMICs.

Here are the suggestions:

1. While the study uses three feature selection methods (Chi-square, LASSO, and SFS), it would be helpful to justify why these specific methods were chosen and discuss whether other feature selection strategies (e.g., mutual information, recursive feature elimination) might yield additional insights or more efficient models.

2. The section describing data cleaning and transformation could benefit from more detail. Specifically, describe the extent and handling of missing data, the criteria for outlier removal, any imputation techniques used, and the distribution of the BMI outcome before and after preprocessing.

3. Since most prior studies used traditional regression, it would be valuable to report, in the Results or Discussion, how the best machine learning model compares in terms of prediction accuracy and feature importance vis-à-vis classical logistic regression.

4. The manuscript concludes with the potential value for public health. Please elaborate with specific examples of how these models could be integrated into health systems or used by practitioners (e.g., screening, resource allocation, or targeted interventions).

**Do you want your identity to be public for this peer review?** For information about this choice, including consent withdrawal, please see our Privacy Policy

Reviewer #1: No

Reviewer #2: No

Reviewer #3: No

Reviewer #4: **Yes:** Qi Wang

---

## [Author Response · Author response to Decision Letter 1]

5 Dec 2025

Dear Editor,

Thank you for your notification regarding our manuscript titled “National data meets AI: Machine learning for predicting overweight/obesity among ever-married Bangladeshi women.” We appreciate the opportunity to revise our work and sincerely thank you and the reviewers for the constructive and insightful feedback.

We have carefully revised the manuscript in response to all comments and suggestions. A detailed, point-by-point response to each comment raised by the Academic Editor and reviewers is provided below.

For your convenience, all modifications within the manuscript have been highlighted using the track-changes feature, and this version has been uploaded as “Revised Manuscript with Track Changes.” We have also uploaded a clean, unmarked version labeled “Manuscript.”

The manuscript has been prepared following all PLOS ONE style and formatting requirements, including the journal’s file-naming guidelines. In accordance with the data and code availability policies, we have shared all relevant code through a publicly accessible GitHub repository and described how to access it in the Data Availability Statement during the submission.

We have also ensured that the abstract in the online submission system matches the abstract in the manuscript. Additionally, although the dataset was initially shared in SPSS (sav) format, some issues were encountered when attempting to open the file. To resolve this, we have also provided the dataset in CSV format.

We believe that these revisions have significantly strengthened the manuscript. We remain grateful to you and the reviewers for your valuable insights and the opportunity to improve our work.

Please let us know if any further changes are required. We look forward to your positive response.

Reviewer #1:

We appreciate the opportunity to revise our work and sincerely thank you for the constructive and insightful feedback. We have carefully revised the manuscript in response to all comments and suggestions. A detailed, point-by-point response to each comment is given by:

Major:

1. Is there any reasons the authors need to use the codes instead of the names of the features? For example, 'age' for feature 'V013'. It is not easy for readers to follow the results part, especially for understanding figure 9 and 10.

-We kept the original variable codes from the BDHS dataset to ensure reproducibility, especially for researchers who may want to replicate the analysis using the same or similar survey datasets. To improve readability, we also presented both the variable code and its descriptive name in the text (e.g., V013 – Age in 5-year groups). This approach maintains reproducibility while also making the results more accessible and easier for readers to understand.

2. It is unclear what parameters were used for SMOTE-ENN and what is the number for underweight/normal and overweight/obese data before and after addressing the class imbalance issue. It would be better if the authors can provide such numbers to make the manuscript transparent and reproducible.

-In the revised manuscript, we have added a detailed description of the SMOTE-ENN parameters used, including sampling strategy and random_state. We also included percentages of samples in each class (underweight/normal vs. overweight/obese), both before and after applying SMOTE-ENN. These details have been added in Lines 160-162, and the updated numbers are also presented in Table 3.

3. Section 3.1 is not very easy to follow, especially from line 340 to 364. The title of this section is 'Results of ML classifier' but these lines to me are like comparing feature selection methods instead of ML classifiers. I think it would be clearer if it can be rephrased like for method xxx, xxx feature selection method gives the best results with accuracy xxx, recall xxx, etc., since the main purpose of the manuscript is to find the best model and interpret the results instead of comparing different feature selecting methods.

-We appreciate your guidance on improving the manuscript's structure and readability. We have reorganized the text to strictly separate the reporting of results from their interpretation. In addition, we have introduced clear subsections within the Results section, specifically "Model Results", "Feature Importance", and “Discussion”, to guide the reader.

4. The interpretation of Figure 10b is not clear. The waterfall plot explains the prediction for single row instead of the whole data (this is why there is gray text before the feature codes showing the value of each feature for this specific data point). The figure 10 legend is a kind of misleading that both panel a and b are about the whole dataset.

-We agree that the waterfall plot may have caused confusion, as it represents the explanation for one specific individual, whereas the SHAP summary plot reflects global feature importance across the entire dataset. To avoid this mismatch and to ensure clearer interpretation, we have removed the waterfall plot from the revised manuscript. Instead, we now present a SHAP summary (violin) plot, which provides a comprehensive global explanation of feature contributions across all observations. This resolves the discrepancy highlighted by the reviewer regarding differences between global and individual feature effects. The figure legend and the corresponding text have been revised to clearly indicate that the results represent global feature importance. Please check Figure 8 in the revised version.

Minor:

1. In line 45-47, the authors compared the obesity percentage of adults in Bangladesh in 1990 and 2022. Since the manuscript is about ever-married Bangladeshi women, is there any statistic (i.e. obesity percentage) of ever-married Bangladeshi women or all Bangladeshi women in 1990 and 2022?

-Data regarding obesity trends are available for the general adult population in Bangladesh when comparing global contexts (Fig 1a). Additionally, we compared statistics for all Bangladeshi women against all Bangladeshi males (Fig 1b). However, because the Bangladesh Demographic and Health Surveys (BDHS) focus exclusively on ever-married women of reproductive age (15-49 years), and historical data for this specific subgroup in 1990 is lacking, our primary analysis focuses on the available BDHS datasets.

2. In the introduction previous study part, the authors mentioned L2 penalty was used for feature selection and algorithm comparison (line 69). Is there any specific reason the authors used L1 (LASSO) penalty in the study?

-Although prior studies (introduction line-80) commonly applied L2 (Ridge) regularization, the present study employed L1 (LASSO) because it performs both coefficient shrinkage and embedded feature selection. By imposing an L1-norm penalty, LASSO forces non-informative or weakly associated predictors to zero, producing a sparse and interpretable set of features. This property was crucial for the current analysis, which aimed to identify the most influential determinants of obesity among ever-married Bangladeshi women. In contrast, L2 regularization only shrinks coefficients without eliminating variables, making it less effective for feature reduction in datasets with correlated socioeconomic predictors. You may please get the discussion under section 2.7.3 in the revised manuscript.

3. In line 516-517 '...suggesting economic status and age strongly influences weight gain.' In this study, the authors only focus on overweight/obesity (static state) instead of weight gain (dynamic process).

-You are correct that our study focuses on the static classification of overweight/obesity rather than the dynamic process of weight gain. We have revised the sentence in lines 611 to accurately reflect this. The updated text now states that economic status and age economic status and age have a significant impact on being overweight and obese, instead of implying weight gain as a dynamic process.

4. In line 507-511, since the data is imbalanced, it would be better to report precision/recall/F1-score instead accuracy when comparing the performance of different models.

-We agree that given the imbalanced nature of the dataset, accuracy alone may not provide a complete picture of model performance. Accordingly, we have revised the manuscript to include Precision, Recall, and F1-score alongside accuracy for all comparison models (e.g., lines 603-605). These metrics provide a more robust assessment of the classifiers' ability to correctly identify the minority class (obesity).

Reviewer #2:

We appreciate the opportunity to revise our work and sincerely thank you for the constructive and insightful feedback. We have carefully revised the manuscript in response to all comments and suggestions. A detailed, point-by-point response to each comment is given by:

1. Most importantly, the author did not include a comparison between the machine learning model and the traditional methods such as logistic regression. A discussion on the advantages and disadvantages of the machine learning model compared to the classic methods in predicting the overweight/underweight is important. For example, in overweight/underweight problems, identifying the more relevant predictors might be more important than the prediction accuracy itself. In that case, the traditional method might be better in terms of explainability.

-We fully agree that a comparison between machine learning models and traditional statistical methods—such as logistic regression—is essential for evaluating both predictive performance and interpretability. In the revised manuscript, we have added a dedicated discussion highlighting the strengths and limitations of both approaches. Specifically, we clarify that while machine learning models generally provided higher predictive accuracy in our analysis, traditional methods like logistic regression offer superior interpretability and clearer identification of key predictors, which can be especially valuable in public health contexts where understanding risk factors is as important as prediction accuracy. Please check the lines 66-72 in the revised manuscript.

Comments:

2. I suggest removing Figure 3 as the information (the number of underweight/overweight women) can be easily conveyed in the main text. This will save room for figures to illustrate more complex results. The information contained in Figure 3 does not contain relevant information. Why only women aged 15-49 were included? What about women who are 50+?

-We appreciate your point regarding Figure 3. Since the information it presents can indeed be conveyed concisely in the main text, we have removed it in the revised manuscript.

Regarding the age range, women aged 15–49 were included because this is the standard reproductive-age group defined in the BDHS dataset. The BDHS 2017-18 only collects full anthropometric measurements (including height and weight) for women within this age range. For women aged 50 and above, BMI data are not consistently available in the dataset, which prevents the inclusion of this group in our analysis. We have clarified this point in the methodology section to avoid potential confusion.

3. As hyperparameter tuning dramatically affects the model performance. How did the author make sure that the tuning parameters are optimized for each specific ML algorithm? The poor performance could be due to the suboptimal hyperparameters.

-In this study, all machine learning algorithms were optimized using a systematic hyperparameter tuning approach with 5-fold cross-validation, rather than relying on default parameter values. Specifically, each model was first trained using its default settings to establish a baseline. Then, we performed an extensive grid search over a predefined range of hyperparameters known to influence model performance. The full search space of initial hyperparameters and the corresponding optimal values obtained for each feature selection method are presented in Table 5.

4. The author should proofread the manuscript. In line 322, the presented accuracy for the machine learning models does not match the result in Table 3. The author should make sure the reported number is consistent between tables and the main context. Two SVMs are mentioned in line 322, and I believe one of these two is a typo. Same problems occurred in line 325. In line 298 “… on the on the respondent’s…”

I’m confused with the data presented in Figure 7. What is the difference between Figure 7 and Figure 5? In addition, some labels at x-axis are missing in Figure 7.

The author should clarify the results and their connections presented in Figure 10 (a) and 10 (b). For example, in figure 10 (a) the SHAP-based feature importance scores from the SVM model highlight that wealth index (V190) is the most significant predictor of obesity, however in figure 10 (b) the contribution from V190 is the lowest 0.01 compared to other features.

-Thank you for carefully identifying these inconsistencies. The discrepancies between the reported accuracy in the main text and those presented in Table 3 (Table 4 in the revised version) occurred because the table was uploaded with incorrect values. The numerical results reported in the manuscript text (lines 385 to 394) are correct. In the revised version, Table 4 has been fully corrected to match the results described in the text, and the duplication of “SVM” has been corrected. Additionally, the typographical error in line 298 (“on the on the respondent’s”) has been corrected.

Additionally, thank you for pointing out the redundancy between Figure 5 (Revised as Figure 4) and Figure 7. Both figures originally presented highly similar information, and Figure 7 did not add any additional analytical value beyond what was already shown in Figure 5. To avoid confusion and improve the clarity of the manuscript, Figure 7 has been removed in the revised version. All relevant results— including the performance of the embedded method—are now fully explained within the text and in Figure 5 (Revised as Figure 4) alone. Instead, we now present only a SHAP summary (violin) plot (Revised Figure 8), which provides a comprehensive global explanation of feature contributions across all observations. This resolves the discrepancy highlighted by the reviewer regarding differences between global and individual feature effects. The figure legend and the corresponding text have been revised to clearly indicate that the results represent global feature importance.

Reviewer #3:

We appreciate the opportunity to revise our work and sincerely thank you for the constructive and insightful feedback. We have carefully revised the manuscript in response to all comments and suggestions. A detailed, point-by-point response to each comment is given by:

Introduction

1.Transitions between global and national-level data discussions and the introduction of machine learning techniques are abrupt.

Recommendation: Include a brief transitional sentence or phrase to smoothly move from global trends to national data.

-A brief transitional sentence or phrase to smoothly move from global trends to regional trends and national data has been added. Please check lines 43-50 in the revised manuscript.

2. The literature review primarily lists prior studies and methods without critically evaluating their strengths, weaknesses, or gaps.

Recommendation: Provide critical evaluation, highlighting limitations of previous methods (e.g., insufficient accuracy, inadequate handling of data imbalance), to justify your study clearly.

- The critical evaluation, highlighting limitations of previous methods (e.g., insufficient accuracy, inadequate handling of data imbalance) has been included in the revised manuscript to justify our study clearly. A few recent studies highlighting their limitations, data quality, and model performance have also been included, and these are stated under the Introduction section.

3. Certain expressions and word choices are informal or less scholarly, e.g.: "So, overweight/obesity is an alarming issue…", inconsistent tense ("were" should be "are").

Recommendation:

Adopt a more formal academic tone. For example: Replace "So" with "Therefore" or "Thus." Use "significant public health concern" instead of "alarming issue."

Consistently use "among" instead of "amid

---

## [Decision Letter · Decision Letter 1]

13 Jan 2026

National data meets AI: Machine learning for predicting overweight/obesity among ever-married Bangladeshi women

PONE-D-25-16485R1

Dear Dr. Biswas,

We’re pleased to inform you that your manuscript has been judged scientifically suitable for publication and will be formally accepted for publication once it meets all outstanding technical requirements.

Kind regards,

Guanghui Liu

Academic Editor

PLOS One

Additional Editor Comments (optional):

Reviewers' comments:

Reviewer's Responses to Questions

**Comments to the Author**

Reviewer #3: All comments have been addressed

Reviewer #4: All comments have been addressed

2. Is the manuscript technically sound, and do the data support the conclusions?

Reviewer #3: Yes

Reviewer #4: Yes

3. Has the statistical analysis been performed appropriately and rigorously?

Reviewer #3: Yes

Reviewer #4: Yes

4. Have the authors made all data underlying the findings in their manuscript fully available?

Reviewer #3: Yes

Reviewer #4: Yes

5. Is the manuscript presented in an intelligible fashion and written in standard English?

Reviewer #3: Yes

Reviewer #4: No

Reviewer #3: The authors have addressed all of my comments appropriately. I recommend the manuscript for publication.

Reviewer #4: (No Response)

**Do you want your identity to be public for this peer review?** For information about this choice, including consent withdrawal, please see our Privacy Policy

Reviewer #3: No

Reviewer #4: **Yes:** Qi Wang

---

## [Editor Report · Acceptance letter]

PONE-D-25-16485R1

PLOS One

Dear Dr. Biswas,

I'm pleased to inform you that your manuscript has been deemed suitable for publication in PLOS One. Congratulations! Your manuscript is now being handed over to our production team.

Kind regards,

on behalf of

Dr. Guanghui Liu

Academic Editor

PLOS One